# The unequal impact of the coronavirus pandemic: Evidence from seventeen developing countries

**Nicolas Bottan** [1]*, **Bridget Hoffmann**[2], **Diego Vera-Cossio**[2]

**1** Policy Analysis and Management, Cornell University, Ithaca, NY, United States of America, **2** Research Department, Inter-American Development Bank, Washington, DC, United States of America

These authors contributed equally to this work.
* nicolas.bottan@cornell.edu

## Abstract

The current coronavirus pandemic is an unprecedented public health challenge that is having a devastating economic impact on households. Using a sample of 230,540 respondents to an online survey from 17 countries in Latin America and the Caribbean, the study shows that the economic impacts are large and unequal: 45 percent of respondents report that a household member has lost their job and, among households owning small businesses, 59 percent of respondents report that a household member has closed their business. Among households with the lowest income prior to the pandemic, 71 percent report that a household member lost their job and 61 percent report that a household member has closed their business. Declines in food security and health are among the disproportionate impacts. The findings provide evidence that the current public health crisis will exacerbate economic inequality and provides some of the first estimates of the impact of the pandemic on the labor market and well-being in developing countries.

## Introduction

Economic inequality is one of the leading economic issues of our era [1–4]. Recent economic downturns, such as the Great Recession of 2008-2009, significantly increased economic inequality [5–7]. Compared to other economic recessions, however, the COVID-19 pandemic is changing economic activity through different channels and on a substantially faster timeline.

To slow the spread of COVID-19, governments have implemented regulations that require social distancing, the closing of non-essential businesses, travel restrictions and, in many cases, stay-at-home orders [8]. Human interactions that drive the economy, such as working together in enclosed areas and enjoying entertainment activities, have been discouraged, restricted, or banned altogether. Residents are complying with these measures, report that they strongly support them [9–11], and actively seek information [12]. Although these measures are necessary for public health, recent evidence from developed countries suggests that they have negative economic impacts in the short-run [13–15] and can potentially deepen the pre-

**Data Availability Statement:** Replication codes and data for this study have been uploaded to the Harvard Dataverse and can be accessed at: https://doi.org/10.7910/DVN/7WX5UU.

**Funding:** This project was funded by the Inter-American Development Bank's Coronavirus research funds (RG-E1700 - Coronavirus survey). The funder provided support in the form of salaries for authors Bridget Hoffmann and Diego Vera-Cossio, and provided funds to cover the cost of recruiting subjects via social media advertising, but did not have any additional role in the study design, data collection and analysis, decision to publish, or preparation of the manuscript. The specific roles of these authors are articulated in the 'author contributions' section."

**Competing interests:** The authors Bridget Hoffmann and Diego A. Vera-Cossio are employees in the Research Department at the Inter-American Development Bank. The author Nicolas Bottan is employed at Cornell University. The authors have no relevant financial or non-financial competing interests to declare. Our affiliations do not alter our adherence to PLOS ONE policies on sharing data and materials.

existing gaps between rich and poor [16]. These negative impacts could be exacerbated in developing countries because firms and the workforce may be more vulnerable due to high levels of informality and weaker governmental capacity to alleviate the pandemic.

This study employs a large-scale online household survey to examine how the COVID-19 pandemic resulted in differential economic impacts for households across the income distribution in Latin America and the Caribbean. The data show large and unequal job losses and business closures, and the effects are strongest for the lowest income households. These negative consequences also translate into declines in food security and support for policies to manage the COVID-19 pandemic. Research on economic inequality is particularly relevant in Latin America and the Caribbean. Although inequality and poverty declined over the most recent decade [17, 18], prior to the pandemic, the region still had the highest income inequality in the world [19], and a large share of citizens were vulnerable to falling back into poverty due to economic shocks [20].

It is important to measure the economic impacts of the COVID-19 pandemic on households in Latin America and the Caribbean. [21], for example, describes the potential disruptive effects of the pandemic across a wide domain of the global economy, including on labor supply and the risks of small business closure and unemployment. This study complements this work by quantifying the short-term implications of the pandemic on job losses and business closures and shows that these impacts further aggravate inequality in the region. Furthermore, [22] describes how macroeconomic spillovers can amplify the adverse economic effects of the pandemic, which would suggest that the estimates obtained are a lower bound. The results presented in this study may be useful to inform pandemic mitigation policy; by indicating where the economic impacts of large-scale lockdowns are particularly large [23].

## Methods

### Design, setting and participants

For the purpose of this study, the questionnaire was mostly standardized across the countries to allow for pooling the responses from all of the countries surveyed. The primary objective of the survey was to measure the economic and well-being impacts that the current pandemic is having on households in Latin America and the Caribbean. For this reason, the questionnaire focused on collecting data on labor market outcomes, financial situation, and social program enrollment. The survey also collected information on hunger, shortages of key goods, and agreement with policies aimed at slowing the spread of COVID-19. A copy of the questionnaire can be found in the Supplementary Information section.

Households in 17 countries were surveyed: 8 South American countries (Chile, Colombia, Bolivia, Ecuador, Guyana, Peru, Suriname, and Uruguay), 4 North and Central American countries (Costa Rica, El Salvador, Mexico, and Panama), and 5 Caribbean countries (Dominican Republic, Bahamas, Barbados, Jamaica, and Trinidad and Tobago). The survey was first launched on March 27, 2020 in Chile, and progressively rolled out to all other countries in our sample by April 17, 2020. With the exception of Costa Rica, data collection continued until April 30, 2020.

### Sampling and validation

The same recruitment methods were followed in all countries. The study recruited participants who were 18 years of age and above using paid advertisements on social media. Participation in the survey was purely voluntary. The advertisements used keywords with broad appeal, such as *fútbol* (soccer) or the names of local celebrities, to avoid selecting participants based on COVID-19 knowledge or interest (details are available in the Supplementary Information

section). For each country, we exclude incomplete surveys, surveys in which the respondent's IP address did not belong to that country, surveys flagged as repeated, surveys with invalid responses, and surveys unreasonable completion time from the sample.

The final sample consists of a total of 230,540 completed responses. S1 Fig in S1 File depicts the geographic coverage of the sample. It shows the number of observations as a share of population (in percent) by sub-national region for each country. The sample achieved broad geographic coverage, with observations in 92 percent of the sub-national regions (see S1 Table in S1 File).

To validate the representativeness of the data, demographic characteristics from the online survey were compared to nationally representative household surveys. Columns (1) and (2) in S3 Table in S1 File shows that although the respondents of the online survey are more educated and more likely to be females, they do not differ substantially in terms of household structure or income levels. Columns (3) and (4) conduct an out-of-sample validation exercise and show that by re-weighting the online survey responses by the inverse probability of being in the nationally representative sample, the differences in demographic characteristics vanish. The Supplementary Information section provides details on the steps taken to estimate weights for the online survey and the validation exercise.

In order to document the economic and well-being impacts of the current pandemic is having on households in Latin America and the Caribbean, most of the analysis presented in this study re-weights observations to achieve national representativeness. These estimates also weight observations according to country population to account for differences in sample size across countries. The exception is Fig 3, which re-weights observations to account for temporal changes in the sample. See the Supplementary Information section for estimation details. All results are robust to not using weights as shown in S2 and S3 Figs and S1 Table in S1 File.

## Statistical analysis

The study presents descriptive statistics for the relevant outcomes aggregated across countries (e.g., rates of job loss and business closure). Details on the analyses conducted for each figure are available in the Supplementary Information. The study further examines how the loss of livelihood relates to changes in household nutrition and policy support by estimating linear regressions. These linear regressions include various controls to isolate time-varying locality shocks and prevent differences in industry sectors from driving results. Refer to the Supplementary Information section for more details on the estimations.

## Results

### COVID-19 and loss of livelihood

The data shows that 45 percent of respondents report that a household member lost a job and, among households owning small family businesses, 58 percent of respondents report that a household member closed their business. The recall period for these questions was randomized between one week, two weeks, and one month. The job-loss rates ranges from 42 percent for a recall period of one week to 47 percent for a recall period of one month. The results suggest that the rates of business closures remain constant at 58 percent across recall periods. Compared to similar statistics in the United States (where 43 percent of small businesses closed [15]), this study finds greater rates of business closure, implying that the economic impacts of the pandemic may be stronger in developing countries.

## Loss of livelihood and inequality

The overall effects obscure highly unequal impacts across income levels prior to the pandemic. Fig 1 shows that after accounting for fixed factors by country, the percentage of households reporting job losses declines monotonically with January 2020 income, prior to the onset of the pandemic. In the case of business closures, the decline is similar though less dramatic. Households reporting income of less than the national monthly minimum wage for January 2020 experienced the largest impacts, with nearly 71 percent reporting that a household member lost their job and 61 percent reporting that a household member closed their business. This contrasts sharply with the impacts reported by respondents with the highest household incomes. Among the highest income respondents, only 14 percent report that a household member lost their job and 54 percent report that a household member closed their business.

One potential explanation for these patterns is that high levels of informality in the region may limit the ability of the most-vulnerable households to maintain their income source. Using the share of self-employed workers as a proxy for the share of informal workers in the labor market, Fig 2 shows that labor market informality is positively correlated with loss of livelihood (job loss or business closure). The slope coefficient suggests that a percentage point increase in the share of self-employed workers in a country increases the likelihood that a respondent lost their livelihood by 0.54 percent (p-value = 0.005) with an R-squared of 42 percent. Because informality rates are high in most developing countries, this result provides a novel explanation for why labor markets in developing countries are particularly hard hit during the crisis.

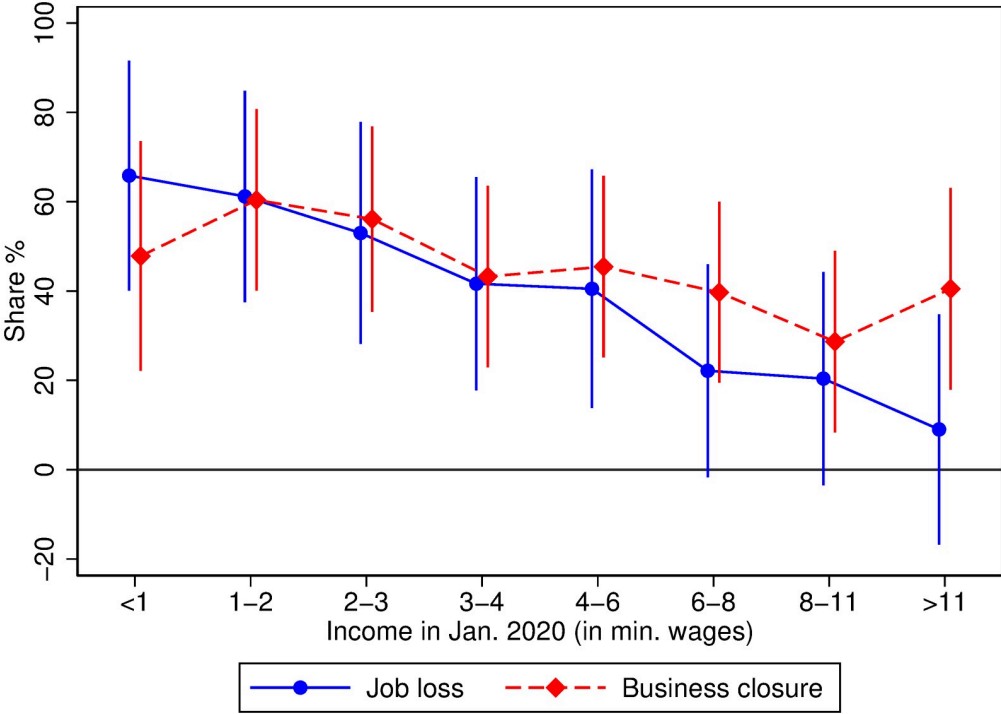

**Fig 1. Higher rates of job loss and business closure among households in the lowest income group.** Point estimates and 95 percent confidence intervals for regressing the labor market outcome on income bin indicators and country fixed effects. Data is weighted using within- and cross-country weights. See Empirical Methods in the Supplementary Information section for details.

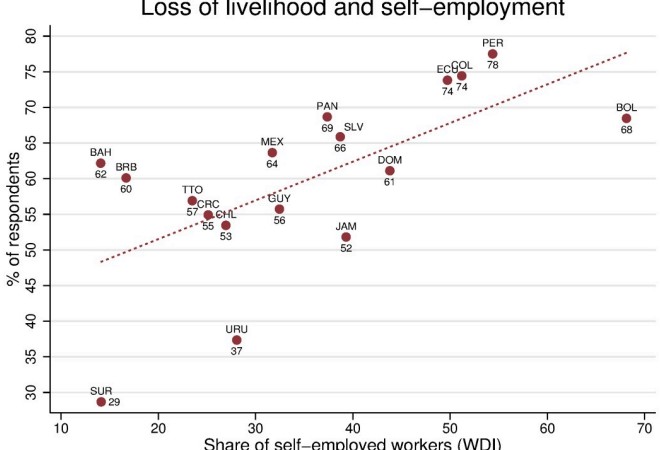

**Fig 2. Higher rates of livelihood loss in countries with higher informality.** Each dot represents the share of respondents who report that a household member lost a job or closed a business. Data is weighted using within- and cross-country weights. See Empirical Methods in the Supplementary Information section for details.

The impacts on job losses and business closures translate into reductions in income. Table 1 reports within-locality changes in outcomes as a response to job losses or business closures. Column 1 of Panel A shows that respondents who report a job loss or business closure are 24 percentage points (p-value < 0.01) more likely to report a reduction in income. Overall, 71 percent of respondents report that they expect their household income in April 2020 to be lower than their January 2020 household income. Thirty-one percent of respondents report household income of less than the national monthly minimum wage for January 2020 and 56% of households report that they expect their household income to be less than the national minimum wage in April 2020. Fig 3 shows that the distribution of household income expected in April 2020 is a leftward shift of the distribution of January 2020 household income. In particular, the share of households with incomes marginally above the national minimum wage declines between January and April 2020, suggesting that many vulnerable households expect to fall into poverty.

## Nutrition and policy support

The data suggest that job losses and business closures lead to reductions in health and food security. Columns 2 and 3 of Panel A of Table 1 shows that households with a job loss or business closure are 13 percentage points (p-value < 0.01) more likely to suffer from hunger and 8 percentage points (p-value < 0.01) more likely to have a less healthy diet relative to their diet prior to the pandemic. These magnitudes represent around a 25 percent increase from the adjusted averages reported by households that report not having lost their livelihood (40 percent).

Column 1 of Panel B of Table 1 shows that households that lost their livelihoods during the crises are 22 percentage points (p-value < 0.01) more likely to receive transfers from relatives or friends. This finding suggests that households cooperate across income levels to smooth the negative economic impacts of the pandemic. Seventy percent of respondents who report household income less than the national minimum wage for January 2020 also report that a household member received a gift or loan from a friend or relative. In contrast, only 26 percent of respondents with the highest incomes in January 2020 report that a household member received a gift or loan from a friend or relative. This pattern reverses for providing a gift or

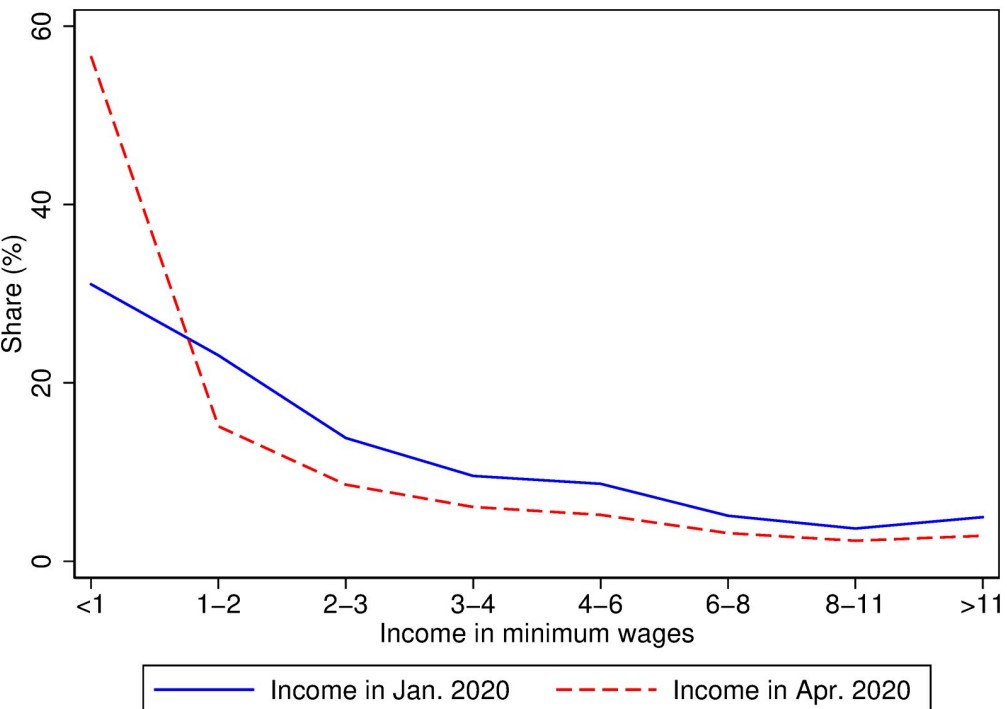

**Fig 3. The share of households in the bottom part of the income distribution is expected to increase.** Shares of households in each income bin for incomes reported for January 2020 and April 2020. Data counts weighted using within- and cross-country weights. See Empirical Methods in the Supplementary Information section for details.

**Table 1. The loss of livelihood during the pandemic is linked to changes in nutrition and policy support.**

| | **Panel A: Impacts on income, food security, and health** | | |
| --- | --- | --- | --- |
| | (1) | (2) | (3) |
| | Decreased income | Went hungry | Eats less healthy |
| Lost job or closed business | 0.241*** | 0.127*** | 0.085*** |
| | (0.008) | (0.008) | (0.008) |
| Observations | 186,058 | 198,190 | 173,956 |
| Adjusted R2 | 0.487 | 0.602 | 0.430 |
| | **Panel B: Impacts on transfers and policy support** | | |
| | (1) | (2) | (3) |
| | Gift/Loan | Gov. Priority | Lockdown ($> =$ month) |
| Lost job or closed business | 0.225*** | -0.027*** | -0.042*** |
| | (0.008) | (0.008) | (0.010) |
| Observations | 198,017 | 196,076 | 125,359 |
| Adjusted R2 | 0.479 | 0.482 | 0.540 |

*$p < 0.1$,

**$p < 0.05$,

***$p < 0.01$.

The table reports regression coefficients capturing the relationship between loss of livelihood (job loss or business closure) and outcomes during the pandemic. Each column reports results of a regression of the dependent variable on an indicator of whether any household member either lost a job or closed a business and a vector of covariates. In addition, all regressions control for locality × day of survey completion fixed effects (18,764), as well as economic-sector fixed effects. Standard errors are clustered at the locality level (3,165). Data is weighted using within- and cross-country weights. See Empirical Methods in the Supplementary Information section for details.

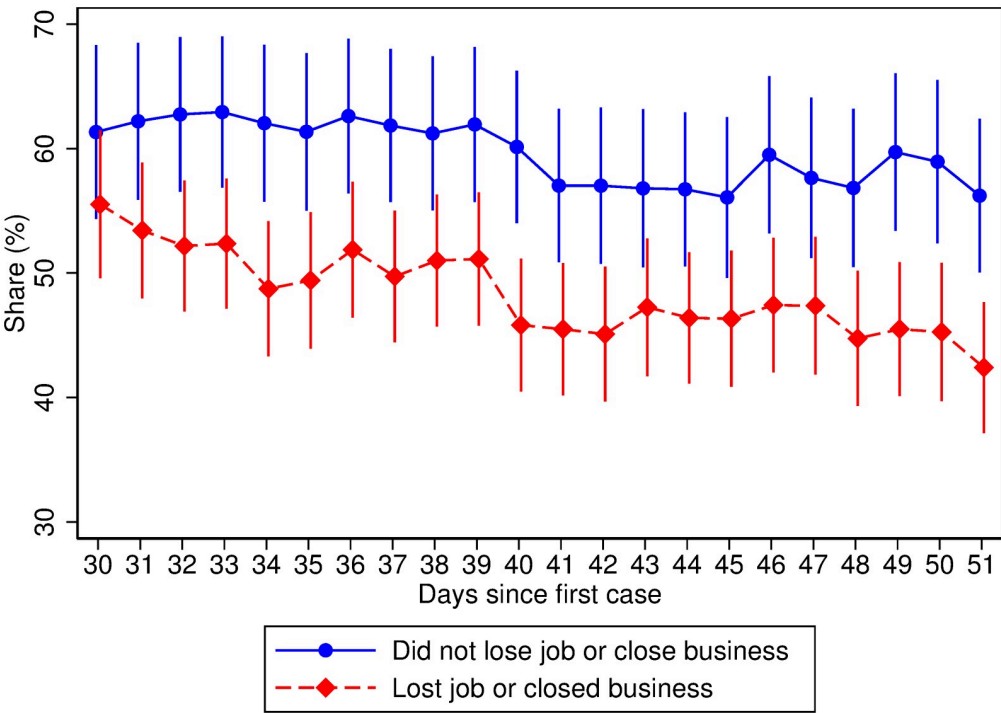

**Fig 4. Support for extending lockdown policies declines more among households that lost their livelihoods.** Point estimates and 95 percent confidence intervals for the share of respondents supporting extending lockdown policies in relation to the number of days since the first COVID-19 case in the country. Data is weighted using within- and cross-country weights. See the Empirical Methods in the Supplementary Information section for details.

loan. Thirty percent of households with January 2020 income less than the national minimum wage also report that a household member provided a gift or loan, compared to 57 percent of households with the highest incomes in January 2020 who report the same. Across all income categories, the receipt of a gift or transfer is concentrated in households that report losing a job or closing a business. This highlights the importance of informal social protection networks as a tool for coping with the negative impacts of the pandemic.

Despite these substantial economic impacts, respondents strongly support measures to slow the spread of the coronavirus in these early stages. Overall, 77 percent of respondents agree with the statement that the top priority of the national government should be to stop the pandemic and 54 percent of respondents think that non-essential businesses should remain closed for an additional month. Although the support for these policies is broad, the support is likely to decrease as more households lose their livelihoods. Column 2 of Panel B from Table 1 shows that the probability of agreeing with the statement that the government's priority should be fighting the pandemic is 2 percentage points lower (p-value < 0.01) among households that experienced a job loss or business closure. Column 3 of Panel B shows that the loss of livelihoods during the pandemic is linked to a 4 percentage-point decline (p-value < 0.01) in the probability of agreeing with the idea of keeping non-essential business closed for an additional month. The support for policies that aim to slow the spread of the coronavirus is thus fragile. Further, Fig 4 shows that, as days go by, the decline in support for keeping businesses closed for an additional month declines faster among households that lose their livelihoods. One important implication is that, without further assistance to impacted households, compliance with mobility restriction policies is likely to decline.

## Discussion

The data suggest two important explanations for the higher vulnerability of households in developing countries with high levels of informality. First, the type of policies that aim to prevent the spread of the virus is likely to affect informal workers more than formal workers (Fig 2). The survey captures data both from countries without enforced mobility-restriction policies or curfews, such as Uruguay, and also from countries with more-stringent, mandatory quarantines and closures of non-essential businesses, as is the case in Bolivia and Peru. Because most of the informal and self-employed workers tend to work in jobs that make them prone to contact with other people (such as those in the retail or services sectors, as opposed to office or industry jobs), the latter set of policies may lead to larger disruptions in labor markets. Indeed, the data indicates that the share of respondents reporting job losses in their households during April (69 percent) is substantially higher in countries with national or local mandatory quarantines, relative to those in countries that did not implement mobility-restriction measures (34 percent) or only curfews (54 percent). See the Supplementary Information sections for a list of countries by type of policy. This finding of smaller economic costs in countries that did not implement national mobility restrictions at the onset of the pandemic than countries with more stringent policies is consistent with evidence from modelling various quarantine regimes gershon2020managing.

Second, differences in the ability to telework could be another reason the negative impacts of the pandemic are concentrated in households with lower incomes. Among respondents that are still employed, the share of respondents that report working from home during the past week increases monotonically with January 2020 household income. Thirty percent of workers from households with incomes below the national minimum wage report working from home, while 76 percent of workers from the highest-income households report working from home.

This study's findings on the inequality of the pandemic's effects across the income distribution suggests that the pandemic may have long-lasting consequences linked to declines in the stock of human capital. Consistent with the results presented in Fig 2, the consequences of the loss of livelihoods on food security are stronger in countries with higher levels of informality (see S5 Table in S1 File). This suggests that the structure of labor markets is not only magnifying exposure to job losses and business closures, but is also magnifying the impacts of the loss of livelihoods on household welfare because informal workers may have less access to formal safety nets.

Taken together, these results show that the negative economic impacts of the COVID-19 pandemic have been concentrated among those who had lower incomes prior to the pandemic. This finding is important from both social and economic perspectives. Inequality is an important social outcome in itself and as well as having important economic implications. Although further research is needed, several studies have found that current inequality is negatively correlated with future economic growth [24, 25], and in particular, inequality driven by the lower tail of the income distribution stunts economic growth [26]. This implies that the unequal economic impacts of this short-term public health pandemic could have long-term implications for economic growth. The results of this study further indicate that country-level rates of informality in labor markets are linked to stronger negative impacts and lower resilience, suggesting that implementing policies to protect informal workers are needed.

## Supporting information

**S1 File.**
(PDF)

## Acknowledgments

We would like to thank Sebastian Espinoza and Maria Paula Medina for superb research assistance. We would like to thank Julián Cristia for his encouragement and advice with the project. We also want to thank German Reyes, Tom Sarrazin, Sebastián Oliva and Pablo Bachelet for their tremendous support in the dissemination of the survey.

## Author Contributions

**Conceptualization:** Nicolas Bottan, Bridget Hoffmann, Diego Vera-Cossio.

**Data curation:** Diego Vera-Cossio.

**Formal analysis:** Nicolas Bottan, Diego Vera-Cossio.

**Funding acquisition:** Bridget Hoffmann, Diego Vera-Cossio.

**Investigation:** Nicolas Bottan, Bridget Hoffmann, Diego Vera-Cossio.

**Methodology:** Nicolas Bottan, Bridget Hoffmann, Diego Vera-Cossio.

**Project administration:** Nicolas Bottan, Bridget Hoffmann, Diego Vera-Cossio.

**Resources:** Nicolas Bottan, Bridget Hoffmann, Diego Vera-Cossio.

**Supervision:** Nicolas Bottan, Bridget Hoffmann, Diego Vera-Cossio.

**Validation:** Nicolas Bottan, Bridget Hoffmann, Diego Vera-Cossio.

**Visualization:** Nicolas Bottan, Bridget Hoffmann, Diego Vera-Cossio.

**Writing – original draft:** Nicolas Bottan, Bridget Hoffmann, Diego Vera-Cossio.

**Writing – review & editing:** Nicolas Bottan, Bridget Hoffmann, Diego Vera-Cossio.

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
