## [Decision Letter · Decision Letter 0]

22 Jul 2020

PONE-D-20-15019

The unequal impacts of the Coronavirus pandemic: Evidence from seventeen developing countries

PLOS ONE

Dear Dr. Bottan,

Thank you for submitting your manuscript to PLOS ONE. After careful consideration, we feel that it has merit but does not fully meet PLOS ONE’s publication criteria as it currently stands. Therefore, we invite you to submit a revised version of the manuscript that addresses the points raised during the review process.

We look forward to receiving your revised manuscript.

Kind regards,

William Joe

Academic Editor

PLOS ONE

Journal Requirements:

"Bridget Hoffmann and Diego Vera-Cossio were joint recipients of the Inter-American Development Bank's Coronavirus research funds (RG-E1700) - www.iadb.org.

We note that one or more of the authors are employed by a commercial company: Inter-American Development Bank.

2.1. Please provide an amended Funding Statement declaring this commercial affiliation, as well as a statement regarding the Role of Funders in your study. If the funding organization did not play a role in the study design, data collection and analysis, decision to publish, or preparation of the manuscript and only provided financial support in the form of authors' salaries and/or research materials, please review your statements relating to the author contributions, and ensure you have specifically and accurately indicated the role(s) that these authors had in your study. You can update author roles in the Author Contributions section of the online submission form.

2.2. Please also provide an updated Competing Interests Statement declaring this commercial affiliation along with any other relevant declarations relating to employment, consultancy, patents, products in development, or marketed products, etc. 

4. We note that Supporting Information includes images of participants in the study. 

5. We note that Figure S1 in your submission contain map images which may be copyrighted. All PLOS content is published under the Creative Commons Attribution License (CC BY 4.0), which means that the manuscript, images, and Supporting Information files will be freely available online, and any third party is permitted to access, download, copy, distribute, and use these materials in any way, even commercially, with proper attribution. For these reasons, we cannot publish previously copyrighted maps or satellite images created using proprietary data, such as Google software (Google Maps, Street View, and Earth). For more information, see our copyright guidelines: http://journals.plos.org/plosone/s/licenses-and-copyright.

5.1.    You may seek permission from the original copyright holder of Figure S1 to publish the content specifically under the CC BY 4.0 license.

5.2.    If you are unable to obtain permission from the original copyright holder to publish these figures under the CC BY 4.0 license or if the copyright holder’s requirements are incompatible with the CC BY 4.0 license, please either i) remove the figure or ii) supply a replacement figure that complies with the CC BY 4.0 license. Please check copyright information on all replacement figures and update the figure caption with source information. If applicable, please specify in the figure caption text when a figure is similar but not identical to the original image and is therefore for illustrative purposes only.

Reviewers' comments:

Reviewer's Responses to Questions

**Comments to the Author**

1. Is the manuscript technically sound, and do the data support the conclusions?

Reviewer #1: Yes

Reviewer #2: Partly

2. Has the statistical analysis been performed appropriately and rigorously? 

Reviewer #1: Yes

Reviewer #2: Yes

3. Have the authors made all data underlying the findings in their manuscript fully available?

Reviewer #1: Yes

Reviewer #2: Yes

4. Is the manuscript presented in an intelligible fashion and written in standard English?

Reviewer #1: No

Reviewer #2: Yes

5. Review Comments to the Author

Reviewer #1: The original research article has a lot of merit and the information presented is of great value. However, presently the format of the manuscript is not in order. A standard research article is composed of 4 main important components; Introduction (background), Methods, Results and a Discussion. The article has a brief introduction which can be further substantiated by mentioning both the global and regional perspectives by referencing recent relevant published literature. some of the suggested articles include:

Gershon D, Lipton A, Levine H. Managing covid-19 pandemic without destructing the economy. arXiv preprint arXiv:2004.10324. 2020 Apr 21.

Kohlscheen E, Mojon B, Rees D. The macroeconomic spillover effects of the pandemic on the global economy. Available at SSRN 3569554. 2020 Apr 6.

Zaman KT, Islam H, Khan AN, Shweta DS, Rahman A, Masud J, Araf Y, Sarkar B, Ullah MA. COVID-19 Pandemic Burden on Global Economy: A Paradigm Shift.

The methods section needs a description of sampling technique. The discussion component is missing and mixed with a extension of results section. the following resource will help in improving quality of reporting the study.

Downes MJ, Brennan ML, Williams HC, Dean RS. Development of a critical appraisal tool to assess the quality of cross-sectional studies (AXIS). BMJ open. 2016 Dec 1;6(12):e011458.

Reviewer #2: 1. Subject to the data availability, sample size selection procedure should be elaborated and clearly written.

2. The method section and sampling procedure should be clearly written to fulfill the aim and objective of the study.

3. The sample selected from each of the 17 countries should be clearly written in the paper.

4. Please provide additional information on how the data was combined and analysed to support the results.

5. Standard English should be used to write the overall paper.

6. As the written English is not clear, so it is difficult to understand the paper properly.

6. PLOS authors have the option to publish the peer review history of their article (what does this mean?). If published, this will include your full peer review and any attached files.

Reviewer #1: No

Reviewer #2: No

---

## [Author Response · Author response to Decision Letter 0]

27 Aug 2020

Please find responses to reviewers in attached file titled 'Response to Reviewers'.

---

## [Decision Letter · Decision Letter 1]

15 Sep 2020

The Unequal Impacts of the Coronavirus Pandemic: Evidence from Seventeen Developing Countries

PONE-D-20-15019R1

Dear Dr. Bottan,

We’re pleased to inform you that your manuscript has been judged scientifically suitable for publication and will be formally accepted for publication once it meets all outstanding technical requirements.

Kind regards,

William Joe

Academic Editor

PLOS ONE

Additional Editor Comments (optional):

Reviewers' comments:

Reviewer's Responses to Questions

**Comments to the Author**

1. If the authors have adequately addressed your comments raised in a previous round of review and you feel that this manuscript is now acceptable for publication, you may indicate that here to bypass the “Comments to the Author” section, enter your conflict of interest statement in the “Confidential to Editor” section, and submit your "Accept" recommendation.

Reviewer #1: All comments have been addressed

2. Is the manuscript technically sound, and do the data support the conclusions?

Reviewer #1: Yes

3. Has the statistical analysis been performed appropriately and rigorously? 

Reviewer #1: Yes

4. Have the authors made all data underlying the findings in their manuscript fully available?

Reviewer #1: Yes

5. Is the manuscript presented in an intelligible fashion and written in standard English?

Reviewer #1: Yes

6. Review Comments to the Author

Reviewer #1: Thank you for addressing all the points mentioned in the previous review. The manuscript is now suitable for publication in the journal.

7. PLOS authors have the option to publish the peer review history of their article (what does this mean?). If published, this will include your full peer review and any attached files.

Reviewer #1: No

---

## [Editor Report · Acceptance letter]

18 Sep 2020

PONE-D-20-15019R1 

The unequal impact of the coronavirus pandemic: Evidence from seventeen developing countries 

Dear Dr. Bottan:

I'm pleased to inform you that your manuscript has been deemed suitable for publication in PLOS ONE. Congratulations! Your manuscript is now with our production department. 

Kind regards, 

on behalf of

Dr. William Joe 

Academic Editor

PLOS ONE